# Wastewater Genomic Surveillance Captures Early Detection of Omicron in Utah

Pooja Gupta,[a] Stefan Liao,[a] Maleea Ezekiel,[a] Nicolle Novak,[a] Alessandro Rossi,[a] Nathan LaCross,[b] Kelly Oakeson,[a] Andreas Rohrwasser[a]

[a]Utah Public Health Laboratory, Utah Department of Health and Human Services, Salt Lake City, Utah, USA
[b]Utah Department of Health and Human Services, Salt Lake City, Utah, USA

**ABSTRACT** Wastewater-based epidemiology has emerged as a powerful public health tool to trace new outbreaks, detect trends in infection, and provide an early warning of COVID-19 community spread. Here, we investigated the spread of SARS-CoV-2 infections across Utah by characterizing lineages and mutations detected in wastewater samples. We sequenced over 1,200 samples from 32 sewersheds collected between November 2021 and March 2022. Wastewater sequencing confirmed the presence of Omicron (B.1.1.529) in Utah in samples collected on November 19, 2021, up to 10 days before its corresponding detection via clinical sequencing. Analysis of diversity of SARS-CoV-2 lineages revealed Delta as the most frequently detected lineage during November 2021 (67.71%), but it started declining in December 2021 with the onset of Omicron (B.1.1529) and its sublineage BA.1 (6.79%). The proportion of Omicron increased to ~58% by January 4, 2022, and completely displaced Delta by February 7, 2022. Wastewater genomic surveillance revealed the presence of Omicron sublineage BA.3, a lineage that was not identified from Utah's clinical surveillance. Interestingly, several Omicron-defining mutations began to appear in early November 2021 and increased in prevalence across sewersheds from December to January, aligning with the surge in clinical cases. Our study highlights the importance of tracking epidemiologically relevant mutations in detecting emerging lineages in the early stages of an outbreak. Wastewater genomic epidemiology provides an unbiased representation of community-wide infection dynamics and is an excellent complementary tool to SARS-CoV-2 clinical surveillance, with the potential of guiding public health action and policy decisions.

**IMPORTANCE** SARS-CoV-2, the virus responsible for the COVID-19 pandemic, has had a significant impact on public health. Global emergence of novel SARS-CoV-2 variants, shift to at-home tests, and reduction in clinical tests demonstrate the need for a reliable and effective surveillance strategy to contain COVID-19 spread. Monitoring of SARS-CoV-2 viruses in wastewater is an effective way to trace new outbreaks, establish baseline levels of infection, and complement clinical surveillance efforts. Wastewater genomic surveillance, in particular, can provide valuable insights into the evolution and spread of SARS-CoV-2 variants. We characterized the diversity of SARS-CoV-2 mutations and lineages using whole-genome sequencing to trace the introduction of lineage B.1.1.519 (Omicron) in Utah. Our data showed that Omicron appeared in Utah on November 19, 2021, up to 10 days prior to its detection in patient samples, indicating that wastewater surveillance provides an early warning signal. Our findings are important from a public health perspective as timely identification of communities with high COVID-19 transmission could help guide public health interventions.

**KEYWORDS** SARS-CoV-2, Omicron, mutation of interest, wastewater surveillance, genomics, public health

Address correspondence to Pooja Gupta, pgupta@utah.gov.

The authors declare no conflict of interest.

Since the beginning of the COVID-19 pandemic, the world has witnessed the evolution of the SARS-CoV-2 virus in real time, with the emergence of new variants and the rapid spread of COVID-19 affecting millions of lives. So far, five variants of concern (VOCs) have been verified by the World Health Organization (WHO): Alpha, Beta, Gamma, Delta, and Omicron (1). Globally, many countries have now lifted COVID-related restrictions, limited vaccine mandates, and transitioned to normality. Thus, there remains an increasing concern over the emergence of novel SARS-CoV-2 variants with increased virulence, transmissibility, and immune escape potential affecting the efficacy of currently available vaccines (2, 3). Over the course of the pandemic, COVID-19 surveillance strategies have relied on testing of symptomatic individuals and their contacts to slow the spread of the pandemic (4, 5). However, symptoms seem to be less severe in the ongoing Omicron wave, with many affected individuals remaining asymptomatic and often not seeking clinical testing, thereby rendering traditional surveillance systems less reliable. Furthermore, increased adoption of at-home testing results in systematic underreporting and less efficient surveillance. Thus, as the pandemic shifts, relying solely on clinical testing may be insufficient for timely detection of outbreaks and emerging variants. To circumvent these challenges, there is an urgent need to incorporate alternative testing strategies to detect low, circulating levels of SARS-CoV-2 variants.

Wastewater surveillance of SARS-CoV-2 is one such approach that can be used to track transmission and spread of new and existing variants across communities (6–10). Although, wastewater-based epidemiology (WBE) has traditionally been used for surveillance of a variety of targets, including poliovirus (11), antimicrobial resistance genes (12), and monitoring illicit drug-use (13), WBE has recently gained traction as a promising epidemiological tool to track SARS-CoV-2 infections. The Centers for Disease Control and Prevention (CDC) developed a national database—the National Wastewater Surveillance System (NWSS)—in late 2020 to monitor the spread of COVID-19 in the United States (14). WBE of SARS-CoV-2 complements existing clinical surveillance as it can capture both symptomatic and asymptomatic cases. As SARS-CoV-2 is known to be shed in human feces post-infection, usually well before symptoms appear (15–17), the viral loads in sewersheds provide a good indicator of SARS-CoV-2 case burden and transmission. Previous studies have found a strong correlation between SARS-CoV-2 concentrations in wastewater and reported COVID-19 cases (18–21).

WBE of SARS-CoV-2 is useful in monitoring infections when clinical testing is limited, expensive or overwhelmed due to case surges, especially during the onset of a new variant. It is also not subject to biases related to health care seeking behavior and willingness to be tested. Previous work has shown that WBE could be essential to conducting long-term, population-wide SARS-CoV-2 surveillance in a rapid and cost-effective manner (22). Importantly, WBE can provide an early warning of rising COVID-19 infections in communities (23, 24) and could be an important tool for guiding public health interventions in combating the ongoing COVID-19 pandemic.

Despite the advantages of WBE for SARS-CoV-2 surveillance, fewer studies implement genomic analysis of wastewater samples. A wastewater sample represents pooled genomic content from multiple individuals and presents a mix of different viral lineages. Additionally, wastewater samples are often degraded and may contain fragmented viral genomes (25). Thus, it is often challenging to confidently assign lineages in a wastewater sample and assess viral genetic diversity. Consequently, early research efforts have focused mainly on measuring virus titers, usually via qPCR, to track trends in COVID-19 infection (6, 19, 20) or digital droplet PCR (ddPCR) to characterize variants of interest (VOI) and VOCs (26, 27). However, PCR-based approaches are limited in their application due to mutations being shared across VOI/VOCs and ongoing evolution of SARS-CoV-2 virus leading to the occurrence of novel mutations. It also requires continual updating and validation of existing methods to avoid primer dropouts (28). As ddPCR targets signature mutations, it may miss amplification of other possible variants that may be epidemiologically relevant. To circumvent these issues, recent studies have adapted the use of next generation sequencing approaches to comprehensively scan the entire SARS-CoV-2 genome for potential mutations

(29–33). This enables detection of unique and functionally important mutations and VOC-defining clusters of mutations. Novel computational approaches are also being developed to parse the different viral lineages in wastewater samples (29, 33, 34). Genomic sequencing of SARS-CoV-2 in wastewater offers a universal and powerful tool to assess the prevalence of circulating SARS-CoV-2 variants and help detect new emerging variants in a population.

In this study, we sequenced wastewater samples collected from 32 sewersheds across Utah (Fig. 1, Table S1). Our goal was to track SARS-CoV-2 spread by characterizing lineages and mutations detected in wastewater communities. We also examined whether lineages detected in wastewater were comparable to lineage diversity observed in clinical samples. We collaborated with the Centre for Genomic Pathogen Surveillance to develop an interactive Micoreact dashboard (35) to visualize our wastewater sequencing data and inform local public health investigations. This internal dashboard provides an intuitive summary of wastewater sequencing results to our state epidemiologists and helps in keeping track of various SARS-CoV-2 mutations and associated lineages over time and across various sampling sites.

## RESULTS

**Viral RNA concentrations.** RT-qPCR analysis was conducted on the raw influent wastewater samples to assess the concentration of viral RNA. Viral RNA concentrations varied widely over the study period from nondetectable levels to a maximum of over 22,000 copies/mL. The maximum concentrations collected from partner facilities also varied by over an order of magnitude. The highest concentrations were observed at the height of the Omicron wave (mid- to late January 2022), while the lowest concentrations were observed shortly thereafter in mid- to late March 2022 (Fig. S1).

**Whole-genome sequencing of SARS-CoV-2 in wastewater samples.** Of the 1,235 samples collected from 32 sewersheds between November 2021 and March 2022, 1,067 had sufficient RNA for whole-genome sequencing. The Illumina DRAGEN COVIDSeq pipeline successfully generated FASTQ files for 1,034 of those samples. We obtained a total of $9.86 \pm 6.08$ million raw reads and retained an average of $8.75 \pm 5.37$ million reads after quality control and read trimming. Of these reads, an average of $62.57 \pm 25.45\%$ mapped to the SARS-CoV-2 reference genome (Fig. S2), with an average coverage of $87.37 \pm 18.94\%$ of the SARS-CoV-2 genome at $1\times$ and $83.59 \pm 19.75\%$ at $10\times$ across all wastewater samples (Fig. S3). Depth of coverage (read depth) varied across the wastewater samples (Fig. S4) and an example of a representative coverage plot is shown in Fig. S5. Further details on number of raw reads, mapping statistics, including mapped reads, reference genome coverage, and depth are provided in Table S2.

**Omicron detection in wastewater preceded clinical surveillance.** We detected a total of 70 different lineages (including sublineages and with frequency greater than 5) from 1,034 consensus sequences following PANGO nomenclature. To assess the diversity of SARS-CoV-2 lineages present in each wastewater sample, we utilized Freyja. Our analysis revealed Delta as the most frequently detected lineage in our data set (frequency, $n = 1,621$, 67.71%) during November, prior to the emergence of Omicron (Fig. 2A, Fig. S6). However, we first detected Omicron (B.1.1.529) in a wastewater sample collected on November 19, 2021, from Hyrum City WWTP (HCWWTP21) at an abundance of 0.1% (Fig. 2C, Table S3). The first clinical case of Omicron in Utah was reported on November 24, 2021, with a specimen collection date of November 18, 2021, though it was travel related. Two additional clinical cases of Omicron were reported on November 29, 2021, but community transmission of Omicron in Utah was observed around mid-December (Fig. 2B and D).

**Wastewater surveillance revealed Omicron sublineage dynamics.** We observed a steady decrease in the proportion of the Delta (B.1.617.2) lineages and the onset of Omicron (B.1.1529) and its sublineage BA.1 ($n = 112$, 6.79%) in wastewater by December. A schematic timeline of the detection of Omicron lineages is shown in Fig. 3. We found BA.1.9 (lineage abundance: 8.35%) from Snyderville Basin Silver Creek (SBWRDSC26) on December 7, 2021, followed by detection of BA.1.17 (lineage abundance: 4.60%) and BA.1.1.11 (lineage abundance: 27.93%) from Central Valley WRF (CVWRF13) on

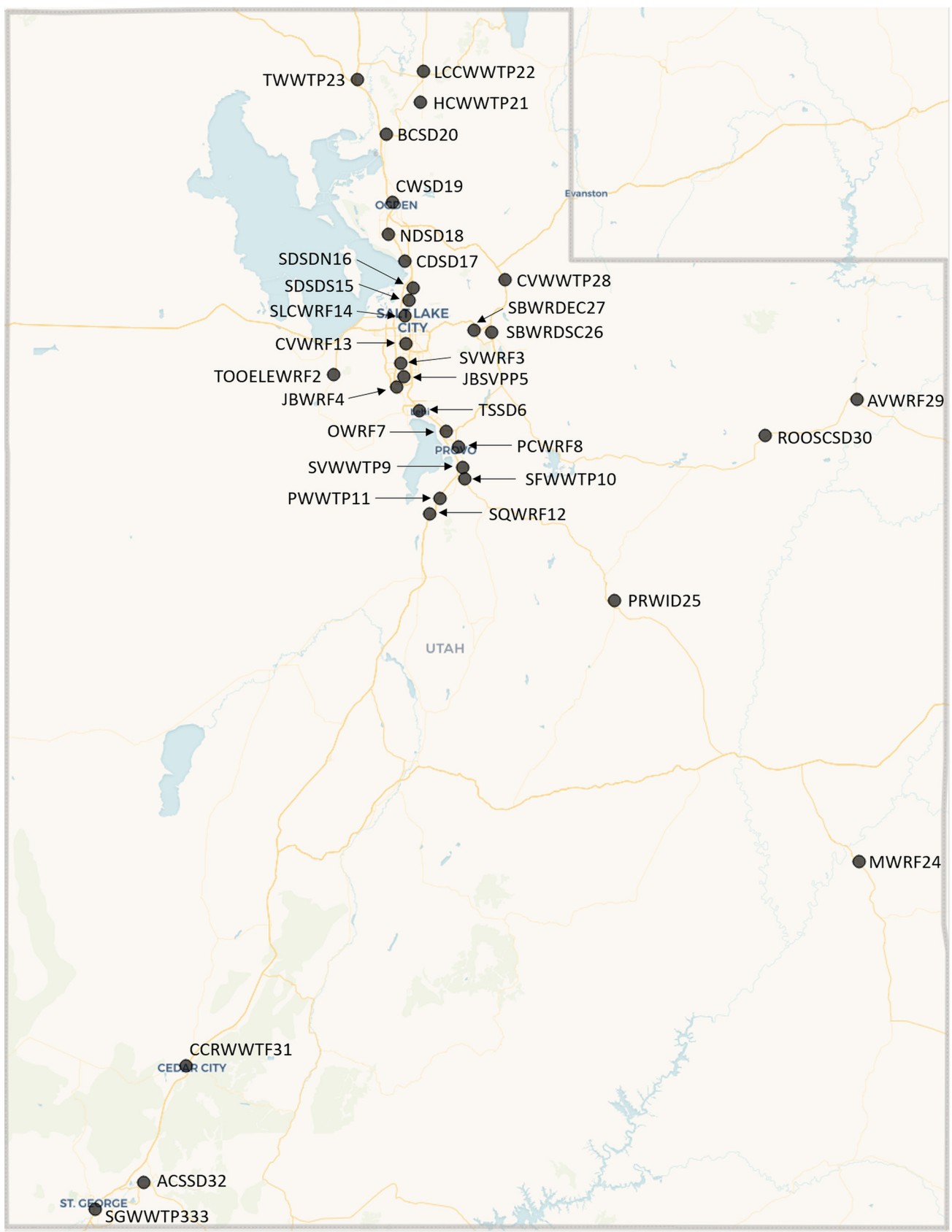

**FIG 1** Geographic map showing the 32 sampled wastewater treatment sites in Utah.

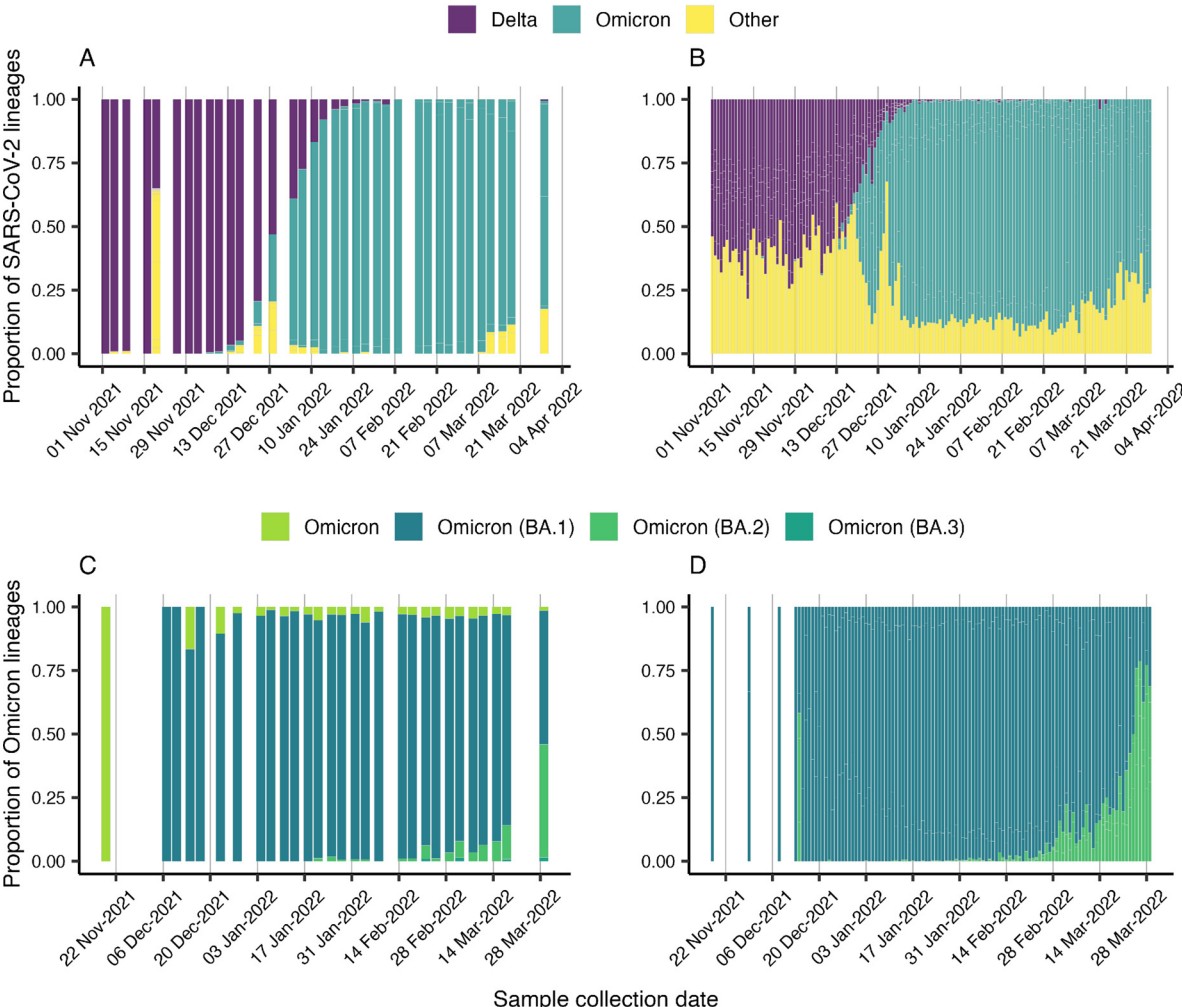

**FIG 2** Proportion of all SARS-CoV-2 lineages in (A) wastewater samples, (B) clinical samples and proportion of Omicron lineages in (C) wastewater samples, and (D) clinical samples from November 2021 to March 2022.

December 10 (Fig. 4, Table S3). By December 14, Omicron had spread to two other communities, Snyderville Basin East Canyon (SBWRDEC27; lineages BA.1.1: 15.8%, BA.1.9: 9.11%) and Brigham City (BCSD20; lineage BA.1.9: 9.82%), along with increased prevalence in Snyderville Basin Silver Creek (SBWRDSC26; lineages B.1.1.529: 9.41%, BA.1.9: 52.75%, and BA.1.1.16: 14.58%). Over the next 2 weeks, several BA.1 sublineages of Omicron had spread to nearly all communities (28/32), including the then common BA.1.15 (Fig. 4). By January 4, prevalence of Omicron in wastewater samples had reached 57.58%, and it completely displaced Delta by February 7, 2022 (Fig. 2A). Between January and March, Omicron accounted for 95% of total lineages in circulation (Fig. 2C), with BA.1 being the most common (*n* = 3,037, 82.93%), followed by BA.2 (*n* = 156, 4.26%) and BA.3 (*n* = 7, 0.19%). Overall, we detected a comparatively rapid shift from the earlier predominant Delta lineage to Omicron in our wastewater data.

Our wastewater genomic surveillance also identified changes in abundance of other Omicron sublineages over time. We first detected the Omicron sublineage BA.2 in our wastewater sequencing data from Ash Creek SSD (ACSSD32) on January 21, 2022 (lineage abundances: BA.2.3, 11.28%, and BA.2, 1.79%; Fig. 4; Table S3). By March 29, the proportion of BA.2 reached about 45% including its sublineages BA.2.1, BA.2.2, BA.2.3, BA.2.4, BA.2.5, BA.2.6, BA.2.7, and BA.2.8. On February 22, 2021, we first detected Omicron sublineage BA.3 (lineage abundance: 6.69%) in Orem WRF (OWRF7). Interestingly, Omicron BA.3 persisted at low abundance in wastewater samples through March, but the BA.3 sublineage

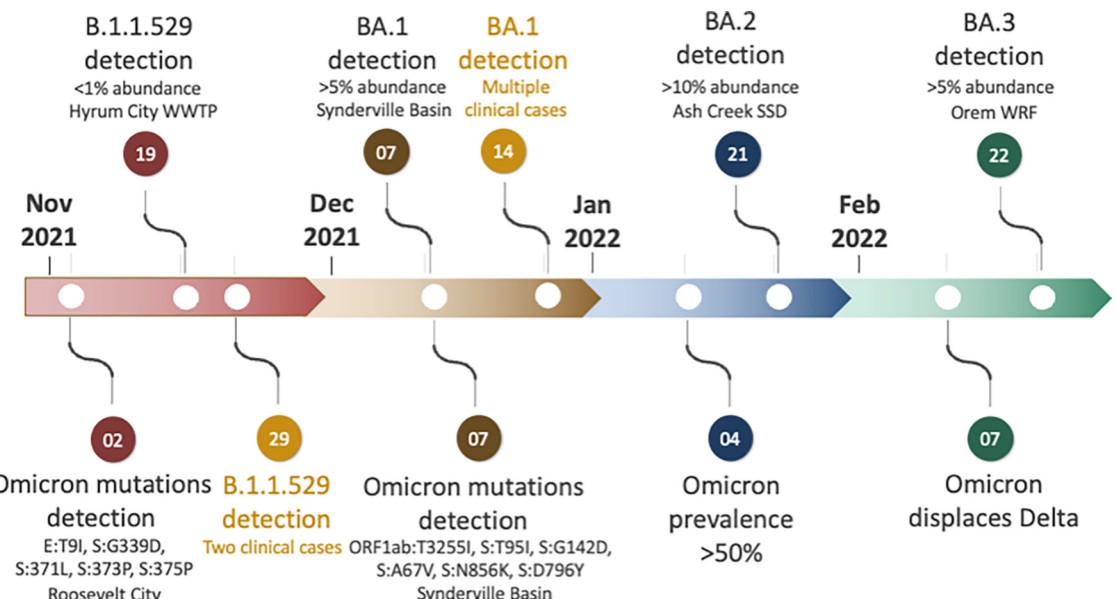

**FIG 3** Timeline of the important events in the detection of lineages and key mutations of Omicron from wastewater sequencing.

has not yet been identified in clinical surveillance. Overall, the diversity of lineages observed in wastewater sequencing data were closely reflected in clinical sequencing (Fig. 2) and resulted in a huge surge in Omicron cases in December to February (Fig. S7). The increased transmission and spike in clinical cases due to Omicron was also evident in wastewater RNA viral loads (Fig. S7) and later also observed in our analysis of the proportion of mutations associated with the Omicron lineage.

**Omicron signature mutations identified in wastewater.** We found S:D614G, ORF1ab:T3255I, S:T478K, S:N969K, and E:T9I as the most common mutations among all wastewater samples (Fig. S8). All five mutations were associated with known VOCs. Notably, all wastewater samples prior to the onset of Omicron (during November 2021) had signature mutations linked to the predominant Delta variant (e.g., S:P681R, S:D950N, T19R), followed by an increase in frequency of mutations shared between Delta and Omicron lineages (e.g., S:L452R, S:T478K; Fig. S9).

An in-depth analysis of mutations from wastewater samples revealed an increase in frequency of several Omicron specific mutations (e.g., ORF1ab:T3255I, S:T95I, S:A67V, S:N856K, S:G142D, allele frequency >0.25) in late December 2021 to January 2022 (Fig. 5). Genomic sequencing for wastewater samples collected from Roosevelt City (ROOSCD30) on November 2nd, 2021 confirmed the presence of five signature Omicron mutations (E:T9I, S:G339D, S:371L, S:373P, S:375P; allele frequency >0.25). We also identified two Spike protein mutations, A67V and T95I (allele frequency 1.00), specific to the Omicron lineage BA.1 from Provo City (PCWRF8) on November 30, 2021. Later, we identified six signature mutations of Omicron (ORF1ab:T3255I, S:T95I, S:G142D, S:A67V, S:N856K, S:D796Y) in Snyderville Basin Silver Creek (SBWRDSC26) sampled on December 7, 2021, supporting the assignment of Omicron BA.1 in our lineage analysis (Fig. 4). We also identified 25 signature mutations of Omicron from Central Valley WRF (CVWRF13) sampled on December 10, 2021, aligning with its lineage assignment as Omicron BA.1 (Fig. 5).

## DISCUSSION

As the SARS-CoV-2 virus continues to spread and evolve rapidly to form new variants with increased virulence or immune escape potential, it is important to monitor and characterize VOI/VOCs. This is especially important now, when pandemic fatigue has become widespread, clinical testing is reduced and more at-home tests are utilized. These factors all contribute toward delays in detecting new emerging lineages when VOCs may be already circulating in communities. In such scenarios, wastewater

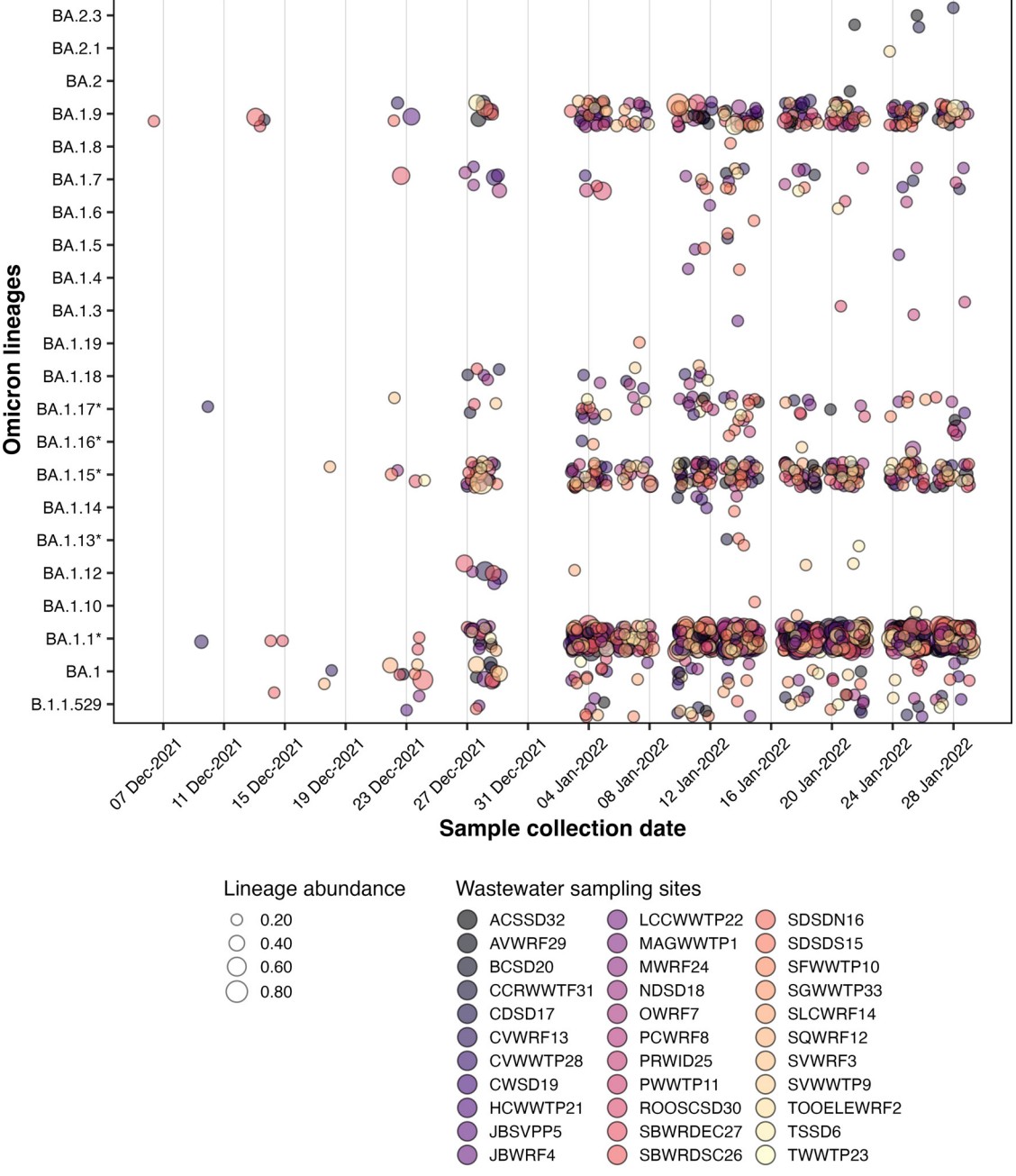

**FIG 4** Abundance of Omicron lineages and its sublineages in each sewershed between December 2021 and January 2022 to highlight the introduction and spread of Omicron in Utah. Circles are color coded by sewersheds and the size of each circle corresponds to the relative abundance of lineages. Lineages with an asterisk include its sublineages.

genomic surveillance could be essential to capture community transmission of COVID in a rapid, cost-effective, and unbiased manner. It can provide a more comprehensive picture of SARS-CoV-2 genomic diversity at the community level than individual level data obtained by clinical sequencing. This will allow timely detection of newly emerging lineages and support public health efforts in mitigating COVID-19 spread (36).

Our wastewater sequencing data provides a first detailed view of the diversity of SARS-CoV-2 lineages circulating between November 2021 and March 2022 in Utah, encompassing the displacement of the Delta by the Omicron lineages. The trends in SARS-CoV-2 RNA levels across sewersheds accurately captured the surge in clinical cases among Utah's population (Fig. S7). Notably, we found that the prevalence and diversity of SARS-CoV-2

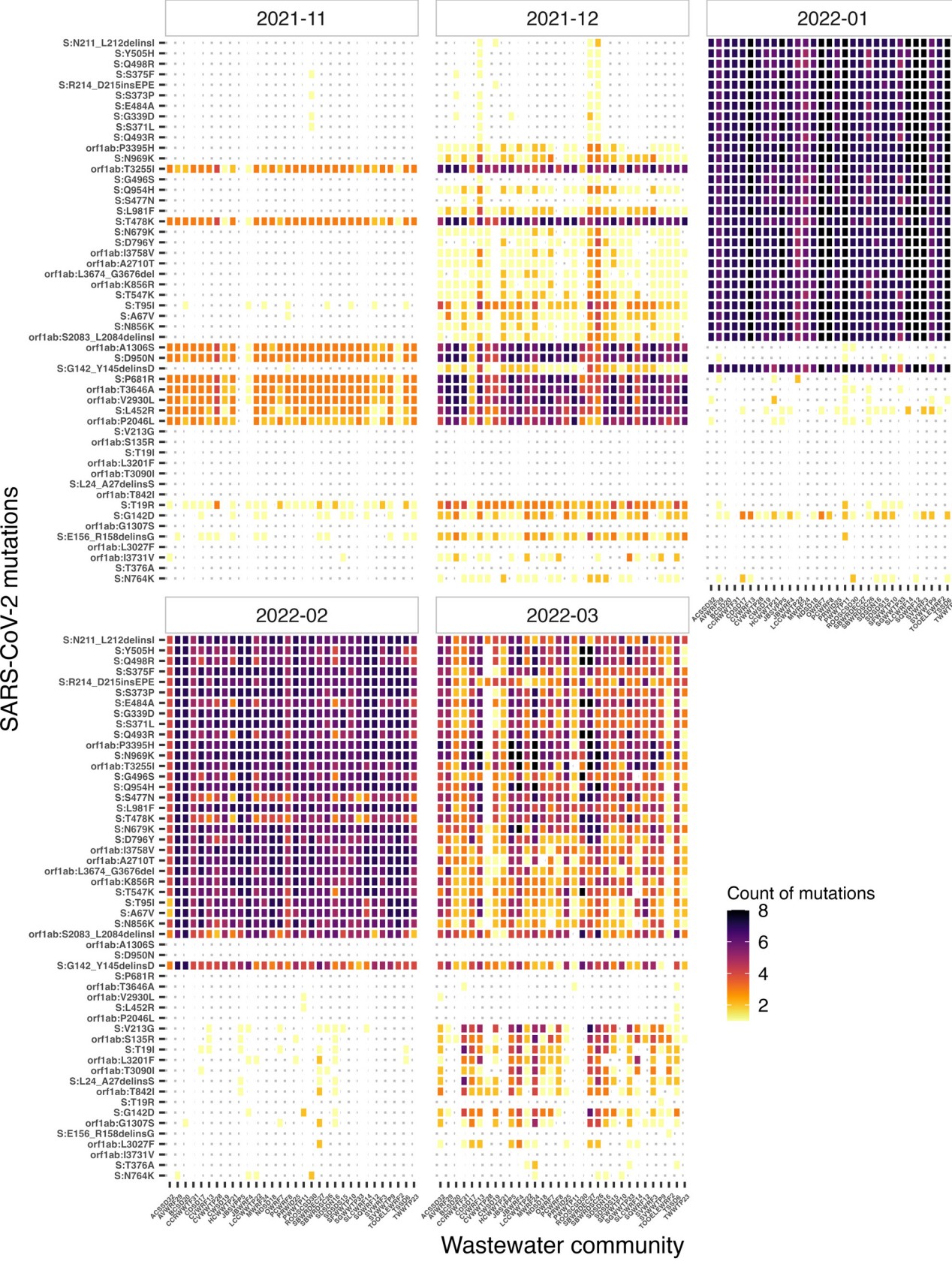

**FIG 5** Heatmap showing the temporal changes in the frequency of SARS-CoV-2 amino acid changes in the Spike and ORF1ab genes, detected in each sewershed from November 2021 to March 2022, encompassing the onset of Omicron. Amino acid changes that were detected only once during the study period are not shown. Colors in the heatmap range from cool yellows/reds representing low frequency to warm blues/violets representing high frequency.

lineages derived from wastewater samples closely aligned with the lineage proportions derived from clinical genomic surveillance. Sequencing of both wastewater samples and clinical samples between December and January revealed a notable increase in the prevalence of Omicron BA.1 lineages and corresponding decrease in the prevalence of Delta lineages (Fig. 2). Omicron BA.1 was the predominant lineage from January through early March, indicating its significant contribution to the overall SARS-CoV-2 burden in the community during the early Omicron phase. This finding is consistent with the global surge in cases observed during the same time period.

One of the main advantages of wastewater surveillance is that it can be used for early detection of emerging lineages and help implement public health interventions to limit variant spread (23, 24). In our study, comparing the collection date of Omicron-positive samples from wastewater with the collection dates of samples from clinical genomic surveillance, we detected Omicron lineages in wastewater genomic surveillance on November 19, 2021, up to 10 days before corresponding detections through genomic clinical surveillance. Notably, the first clinical case was detected in Utah on November 18, 2021, but was a result of someone with international travel history to a region with high community transmission of Omicron. Two additional clinical cases were identified on November 29, 2021. Subsequent increases in abundance of Omicron lineages in wastewater on December 7, 2021, were followed by an increase in the number of clinical cases by December 14, supporting the week ahead early warning signal detected in wastewater data. The lead time between the detection of a new lineage in wastewater compared to clinical surveillance is still difficult to estimate in advance and has been shown to vary from 5 to 14 days (19, 23, 24, 37). The lead time can potentially be influenced by environmental factors—temperature, seasons (38), biological factors such as the shedding rates associated with a lineage (39), severity of infection, and whether an individual is symptomatic or asymptomatic (37) when contributing to a particular catchment.

In our study, analysis of wastewater samples by Freyja, which provides relative abundance of multiple lineages in a sample, revealed detection of Omicron (B.1.1.529) from a Hyrum City (HCWWTP) sample collected on November 19, 2021. However, based on Pangolin lineage assignment on consensus sequences, we did not detect the Omicron lineage (BA.1.1) until December 13, 2021, in Snyderville Basin Silver Creek sample. This highlights the uncertainty in lineage assignment based on consensus sequences derived from wastewater samples, which can contain multiple viral lineages. Pangolin lineage assignment tools are perhaps better at identifying the predominant viral lineage in a community but unlikely to represent the true lineage diversity in a wastewater sample. Thus, the use of complementary analysis tools, such as Freyja, which provides an abundance of multiple viral lineages in a sample, is important to gain a comprehensive understanding of the viral landscape in a community for effective SARS-CoV-2 surveillance.

Throughout the duration of our study encompassing the peak transmission of Omicron, wastewater sequencing identified three sublineages of Omicron: BA.1, BA.2, and BA.3. Omicron sublineage BA.1 was detected more frequently during our study compared to BA.2 across all sampled communities. By the end of March, as Omicron spread progressed, we observed an increased prevalence of BA.2 (45% prevalence on March 29), similar to its prevalence in other states in the United States. While BA.2 was still dominant across communities, we first detected BA.3 from Orem WRF on February 22, 2021. Surprisingly, BA.3 has not been detected in clinical surveillance in Utah. This could possibly be due to its low prevalence as was observed in our wastewater data (Fig. 2C) and has been observed throughout the US (<0.5% prevalence, last checked September 19, 2022, on outbreak.info; [40]), suggesting its cryptic circulation in Utah's wastewater communities. These findings are in agreement with previous studies that highlight the utility of wastewater surveillance in detection of cryptic lineages, or novel SARS-CoV-2 lineages that are rarely observed in humans (7, 33).

Interestingly, the shift from Delta to Omicron was also quite clear from our analysis of SARS-CoV-2 mutations in wastewater. We found predominance of Delta mutations in November-early December, followed by an increase in the number of mutations

common between Delta and Omicron lineages and then an increase in Omicron defining mutations (Fig. 5 and Fig. S9). Thus, our ongoing wastewater genomic surveillance allowed us to detect Omicron lineage early even when based on its signature mutations (e.g., S:A67V, S:N856K, S:N969K, E:T9I). Our study demonstrates that early detection of a lineage is feasible based on signature mutations, but caution is required when interpreting these data, particularly during the early stages of the emergence of a new lineage. The low frequencies of signature mutations associated with a new lineage can make it challenging to distinguish the actual signal from noise in the sequencing data. Hence, combining information from multiple signature mutations is crucial to increase the accuracy of lineage detection.

To identify novel SARS-CoV-2 lineages confidently in wastewater, a comprehensive and well-curated lineage database is also essential. Novel lineages that have recently diverged from their ancestral lineage and have not yet been curated in existing databases can be difficult to identify via wastewater sequencing, but analyzing key mutations from a clinically important region of the SARS-CoV-2 genome such as the Spike gene could help identify potential new lineages in wastewater. Some studies discovered rare, cryptic lineages in wastewater that did not match any known lineages and contained several rare mutations that had not been identified in routine clinical surveillance but found to be associated with the circulating VOCs (7, 41). However, these studies sequenced a small portion of the virus's genome called the RBD (Receptor Binding Domain) region of the Spike protein, which may provide limited information and may miss critical mutations for accurate lineage characterization. Furthermore, mutations in other regions of the virus could also contribute to immune escape, transmissibility, and pathogenicity. Thus, whole-genome sequencing is a better approach for routine monitoring of mutations of concern in wastewater, providing a comprehensive view of the virus's evolution and diversity, and identifying recombination events and the emergence of new mutations.

Overall, our study illustrates that genomic sequencing of wastewater samples can be used to detect, monitor, and provide insights into the diversity of SARS-CoV-2 variants. Of note is the cost-effectiveness of wastewater sequencing as a pooled community sample can be analyzed to monitor disease spread. Wastewater-based genomic epidemiology can be used to track the prevalence of SARS-CoV-2 variants in near real-time, often earlier than information obtained by clinical sample sequencing. Early detection of emerging lineages can provide the time needed to allocate more testing resources to communities showing early detection of an emerging variant and implement public health interventions to limit variant spread. As we move beyond Omicron and new variants emerge, wastewater-based surveillance will be a valuable tool in the timely identification of novel lineages, complementing clinical sequencing and aiding in public health decision-making. While wastewater surveillance has proven to be an essential tool for SARS-CoV-2 genomic surveillance, its noninvasive and cost-effective nature also makes it an attractive option for monitoring public health on a large scale and there is potential to expand its application to detect other emerging infectious diseases. Integrating wastewater surveillance into public health surveillance systems could enhance detection and response to emerging infectious diseases, mitigating their impact on public health and the economy.

## MATERIALS AND METHODS

**Sample collection.** Sewage samples were collected twice per week at 32 wastewater facilities between November 2021 and March 2022 ($n = 1,235$; Fig. 1; Table S1). All samples were raw influent wastewater collected from municipal treatment systems. Estimated resident populations varied widely, from a low of 1,322 to a high of 515,494. While the specific type of sample collected was dependent on sampling equipment available at each partner facility, all but three facilities provided nominally 24-h time- or flow-weighted composite samples (mean duration of 23.9 h, range of $16 - 28.3$ h). The remaining three facilities provided either grab samples (Roosevelt City SD) or 6-h manual composite samples (Snyderville Basin East Canyon and Snyderville Basin Silver Creek; four grab samples collected every 2 h over a total of 6 h, then mixed). Samples were stored in a refrigerator or on ice and transported to the Utah Public Health Laboratory (UPHL) within 24 h.

**Sample processing.** Twenty mL of raw influent wastewater sample was processed following the steps outlined in the Promega Maxwell RSC Enviro Total Nucleic Acid kit (Promega Corp.). Briefly, the samples were lysed, the solids were separated, and the samples were buffered. Total nucleic acids were captured on the Promega PureYield Midicolumn, purified and eluted to 600 $\mu$L. The 600 $\mu$L concentrate was then extracted by Promega Maxwell RSC 48 automated extraction and stored at $-80°C$ until RT-qPCR analysis.

**RT-qPCR assay.** We conducted RT-qPCR analysis using 20 $\mu$L reactions consisting of 5 $\mu$L template, 7 $\mu$L nuclease-free water, 5 $\mu$L TaqPath 1-Step RT-qPCR Master Mix ($4\times$), and 1.5 $\mu$L of each N1 and N2 primer/probe mix from the 2019-nCoV CDC EUA kit (42) at a final concentration of 1.125 $\mu$M. We performed analysis using the Applied Biosystems 7500 Fast (Dx) real-time PCR system with a cycling condition of 2 min at 95°C followed by 45 cycles of 3 s at 95°C and 30 s at 55°C. SARS-CoV-2 RNA quantification was conducted using a 5-point standard curve generated from serial dilutions of the CDC EUA 2019-nCoV positive control, ranging from 20 copies/rxn to $1 \times 10^5$ copies/rxn. The required standard curve criteria were an efficiency between 80 and 120% and $R^2$ greater than 0.98. We monitored the stability, method performance, and quantitative consistency by recording the $C_T$ of the $1 \times 10^5$ copies/rxn standard well longitudinally at a predetermined threshold.

**Whole-genome Sequencing.** We sequenced the SARS-CoV-2 genome using amplicon-based sequencing following the Illumina COVIDSeq RUO protocol (Illumina, Inc., San Diego, CA, USA). Library preparation was automated via Tecan liquid handlers and pooled, amplified libraries were loaded onto the S4-flow cell following NovaSeq workflow as per manufacturer's instructions (Illumina Inc.). Dual indexed single-end sequencing with a 75 bp read length was performed on the NovaSeq 6000 platform.

**Bioinformatics analysis.** After sequencing, we generated FASTQ files from the raw data using the DRAGEN COVIDSeq Test Pipeline (Illumina Inc.) on the Illumina DRAGEN Bio-IT platform. We used viralrecon (v.2.4.1) to analyze the sequencing data and detect SARS-CoV-2 variants (43). Briefly, we performed data QC, trim, and quality filtering on raw FASTQ files using fastp. We discarded reads with an average Phred quality score below 30 and length shorter than 50 bp. We mapped the trimmed, human-filtered reads to the SARS-CoV-2 (Wuhan-Hu-1 reference, RefSeq ID: NC_045512.2) genome using bowtie (v.2.4.4) (44). We soft-clipped the ARTIC primers using iVar (v.1.3.1) (45) using default settings. Subsequently, we performed variant calling using iVar and generated consensus sequences using BCFtools (v.1.14) (46). We retained variants with a minimum base quality of 20, allele frequency of 0.25 and coverage of 10 reads. We annotated variants with SnpEff (v.5.0e) (47) and SnpSift (v.4.3) (48). All filtered variants were then compared against a manually curated list of known and epidemiologically important SARS-CoV-2 variants (including VOCs and VOIs) built using the outbreakinfo R package (49).

We used Pangolin COVID-19 Lineage Assignment tool (v.4.0) to detect potential SARS-CoV-2 lineages from consensus sequences (50). We obtained a summary of raw reads, sample QC, alignment, and variant calling metrics via MultiQC (v.1.11) (51). Unlike a clinical sample, each wastewater sample can contain multiple different SARS-CoV-2 virus lineages shed by many infected individuals. We used Freyja (v1.3.8) (33) to delineate and ascertain the relative proportion of different SARS-CoV-2 lineages in each wastewater sample. To compare SARS-CoV-2 lineage diversity observed in wastewater with clinical samples, we extracted prevalence of different SARS-CoV-2 lineages from clinical surveillance in Utah using the outbreakinfo R package (49).

**Data availability.** Raw sequencing data have been deposited on the NCBI Sequence Read Archive under accession number PRJNA812604. The code used for analysis and visualization in the manuscript can be accessed at https://github.com/poojasgupta/wastewater-covidseq.

## SUPPLEMENTAL MATERIAL

Supplemental material is available online only.

**SUPPLEMENTAL FILE 1**, XLSX file, 0.01 MB.

**SUPPLEMENTAL FILE 2**, CSV file, 0.2 MB.

**SUPPLEMENTAL FILE 3**, CSV file, 0.9 MB.

**SUPPLEMENTAL FILE 4**, DOCX file, 7.4 MB.

## ACKNOWLEDGMENTS

This work was supported by the Centers for Disease Control and Prevention's (CDC) Epidemiology and Laboratory Capacity for Prevention and Control of Emerging Infectious Diseases (ELC) Enhanced Detection Cooperative Agreement. We thank our partner wastewater treatment facilities across the State of Utah who collected and sent us the wastewater samples. We also thank the UPHL NGS Sequencing Team for conducting the sequencing of wastewater samples. The graphical abstract was created with BioRender.com.

Pooja Gupta: conceptualization, methodology, formal analysis, writing–original draft, writing–review and editing, visualization. Stefan Liao: methodology, investigation, writing–review and editing. Maleea Ezekiel: methodology, investigation, writing–review and editing. Nicolle Novak: methodology, investigation, writing–review and editing. Alessandro Rossi: funding acquisition, writing–review and editing. Nathan LaCross: funding acquisition,

investigation, supervision, project administration, methodology, writing–review and editing. Kelly Oakeson: funding acquisition, conceptualization, supervision, project administration, methodology, writing–review and editing. Andreas Rohrwasser: funding acquisition, writing–review and editing.

The authors have no conflicts of interest to declare.

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
