## [Reviewer comments · Microbiology Spectrum]

Microbiology Spectrum

Wastewater genomic surveillance captures early detection of Omicron in Utah

Pooja Gupta, Stefan Liao, Maleea Ezekiel, Nicolle Novak, Alessandro Rossi, Nathan LaCross, Kelly Oakeson, and Andreas Rohrwasser

Corresponding Author(s): Pooja Gupta, Utah Department of Health

Review Timeline:

Submission Date:	January 25, 2023
Editorial Decision:	February 5, 2023
Revision Received:	April 6, 2023
Accepted:	April 12, 2023

Editor: Abimbola Kolawole

Reviewer(s): Disclosure of reviewer identity is with reference to reviewer comments included in decision letter(s). The following individuals involved in review of your submission have agreed to reveal their identity: Jonathan Daniel Hulse (Reviewer #5)

Transaction Report:

DOI: <https://doi.org/10.1128/spectrum.00391-23>

February 5, 2023

Dr. Pooja Gupta
Utah Department of Health
4431 South 2700 West
Salt Lake city, Utah 84129

Re: Spectrum00391-23 (Wastewater genomic surveillance captures early detection of Omicron in Utah)

Dear Dr. Pooja Gupta:

Link Not Available

Sincerely,

Abimbola Kolawole

Journals Department
Reviewer comments:

Reviewer #2 (Comments for the Author):

This manuscript aimed to track SARS-CoV-2 variants/lineages changes in communities in Utah using whole genome sequencing of wastewater samples. The authors evaluated more than 1,000 samples across 32 sewersheds to track the presence of Omicron sublineages in local communities and compared the timing with reported clinical cases. For sequencing methods, the Artic scheme-based Illumina COVID-seq was used, resulting an overall genome breadth of coverage at 87.37{plus minus}18.94% at 1X depth across all samples. Overall, the manuscript is well structured and data analysis is well performed, the conclusions are also clearly supported by the data. There are some minor issues to address, mainly regarding methods and data visualization. Also, please check through all references to make sure the latest version is cited (e.g., ref 43).

Comments for the authors' consideration:

1. Line 55, Line 59-61, Line 78-79, and Line 100: Please add corresponding references to these sentences.
2. Line 76: Check citation style.
3. Line 138-141: Please provide detailed methods for the N1 and N2 RT-qPCR, e.g., reaction system volumes and amplification program.
4. Line 141-142: Please provide detailed information regarding the N1 and N2 standard curves, i.e., slope, y-intercept and efficiency.
5. Line 162: Does the author have any results from the SnpEff annotation? If not, I suggest removing this sentence.
6. Line 185-186: What is the average depth across all samples?
7. Line 187: One question I have is that how the author decided lineages and the corresponding relative abundances? As the author mentioned in the discussion, there is a disagreement in the lineages identified through the consensus method (Pangolin) versus direct mapping (Freyja). Yet in this paragraph it looks like the author adopted lineages from the Pango approach and the relative abundance from Freyja? Please clarify.
8. Figure 190: Looks like the order of Figure 3 and Figure 2 is reversed, please check and clarify.
9. What is the "other" lineage (yellow bar) in Figure 3a, date 11-15-21? This is very eye-catching in the figure and needs to be described in some way.
10. Line 192: The 11-19 sample in Figure 3b looks like a 100% Omicron (light green bar). Please clarify.
11. Line 203: Figure 4 is very busy. Consider showing only the important or mentioned lineages in the text (the ones of higher relative abundances) and keep the raw data of all Omicron lineage in the supplemental materials, e.g., a supplemental table.
12. Line 222-223: This sentence needs to be clarified. It sounds like the author is talking about the pattern of BA.3 while it actually means the pattern of all Omicron sublineages.
13. Line 237: For Figure 5, I cannot see the details due to the low resolution. Consider better annotation some of the mutations on the y-axis, e.g., color coding these lineage-specific mutations.
14. Line 240: Are all these omicron specific mutations at similar allele frequencies?
15. Line 307-309 and Line 327-329: These studies used the RBD region only sequencing method. To strengthen the study's importance, I suggest the author adding in the discussion, or introduction, the advantage of sequencing whole genome in comparison to sequence a selective sub-region.

Reviewer #3 (Comments for the Author):

1. Wastewater-based epidemiology (WBE) has become very popular and powerful as an unbiased, passive, recruitment-independent tool to monitor emergence and re-emergence of infectious diseases in post-covid era. The most recent successful example is the sustained detection of polio virus sequence in London sewage that prompted urgent vaccination advice to susceptible population. While similar studies have been reported in many cities, there are a few aspects that remain understudied. One such aspect is the effect of catchment population on WBE performance. Intuitively, a large catchment size can reduce the number of sampling points to cover the whole city population but will likely result in reduced sensitivity as the source viral RNA will become more diluted before reaching sewage treatment plants. In this study, the catchment population size of each sewage sampling site varied widely, from as low as 1,322 to a high of 515,494. It would be interesting to know if there was any association between WBE performance with catchment population size? Were early warnings (i.e., detection of viral sequences in sewage before clinical case) more commonly contributed by less populated sampling sites? Such data would add values to the study if available and could advise on attributes that define the most cost-effective WBE design study.

2. Short single end 75bp mNGS was performed. Were longer contigs generated by de novo assembly before feeding to variant mapping and calling?

3. Methodological details on the collection of virus-related information from clinical settings appears to be missing.

Reviewer #5 (Comments for the Author):

This is one of the most fluid and concise papers that deals with wastewater identification of viral components. This manuscript is well written and is going to add to the growing body of work surrounding SARS-CoV-2. This paper presents the results in a very readable format, and the data is presented in an effective format. The impact of this study will help public health professionals identify potential new SARS-CoV-2 variants, and help plan for potential future pandemics.

Staff Comments:

Preparing Revision Guidelines

Please return the manuscript within 60 days; if you cannot complete the modification within this time period, please contact me. If you do not wish to modify the manuscript and prefer to submit it to another journal, please notify me of your decision immediately so that the manuscript may be formally withdrawn from consideration by Microbiology Spectrum.

This manuscript aimed to track SARS-CoV-2 variants/lineages changes in communities in Utah using whole genome sequencing of wastewater samples. The authors evaluated more than 1,000 samples across 32 sewersheds to track the presence of Omicron sublineages in local communities and compared the timing with reported clinical cases. For sequencing methods, the ARTIC scheme-based Illumina COVID-seq was used, resulting an overall genome breadth of coverage at $87.37 \pm 18.94\%$ at 1X depth across all samples. Overall, the manuscript is well structured and data analysis is well performed, the conclusions are also clearly supported by the data. There are some minor issues to address, mainly regarding methods and data visualization. Also, please check through all references to make sure the latest version is cited (e.g., ref 43).

Comments for the authors' consideration:

1. Line 55, Line 59-61, Line 78-79, and Line 100: Please add corresponding references to these sentences.
2. Line 76: Check citation style.
3. Line 138-141: Please provide detailed methods for the N1 and N2 RT-qPCR, e.g., reaction system volumes and amplification program.
4. Line 141-142: Please provide detailed information regarding the N1 and N2 standard curves, i.e., slope, y-intercept and efficiency.
5. Line 162: Does the author have any results from the SnpEff annotation? If not, I suggest removing this sentence.
6. Line 185-186: What is the average depth across all samples?
7. Line 187: One question I have is that how the author decided lineages and the corresponding relative abundances? As the author mentioned in the discussion, there is a disagreement in the lineages identified through the consensus method (Pangolin) versus direct mapping (Freyja). Yet in this paragraph it looks like the author adopted lineages from the Pango approach and the relative abundance from Freyja? Please clarify.
8. Figure 190: Looks like the order of Figure 3 and Figure 2 is reversed, please check and clarify.
9. What is the "other" lineage (yellow bar) in Figure 3a, date 11-15-21? This is very eye-catching in the figure and needs to be described in some way.
10. Line 192: The 11-19 sample in Figure 3b looks like a 100% Omicron (light green bar). Please clarify.

11. Line 203: Figure 4 is very busy. Consider showing only the important or mentioned lineages in the text (the ones of higher relative abundances) and keep the raw data of all Omicron lineage in the supplemental materials, e.g., a supplemental table.
12. Line 222-223: This sentence needs to be clarified. It sounds like the author is talking about the pattern of BA.3 while it actually means the pattern of all Omicron sublineages.
13. Line 237: For Figure 5, I cannot see the details due to the low resolution. Consider better annotation some of the mutations on the y-axis, e.g., color coding these lineage-specific mutations.
14. Line 240: Are all these omicron specific mutations at similar allele frequencies?
15. Line 307-309 and Line 327-329: These studies used the RBD region only sequencing method. To strengthen the study's importance, I suggest the author adding in the discussion, or introduction, the advantage of sequencing whole genome in comparison to sequence a selective sub-region.

We thank the Reviewers for the positive comments and valuable suggestions for improving our manuscript. We have revised the manuscript as per the recommendations made by the Reviewers and hope the revised text, figures and comments sufficiently addresses all concerns. Please see our point-by-point response (in blue) to the specific suggestions made by the Reviewers below, along with the corresponding line numbers from the revised manuscript.

Reviewer comments:

Reviewer #2 (Comments for the Author):

This manuscript aimed to track SARS-CoV-2 variants/lineages changes in communities in Utah using whole genome sequencing of wastewater samples. The authors evaluated more than 1,000 samples across 32 sewersheds to track the presence of Omicron sublineages in local communities and compared the timing with reported clinical cases. For sequencing methods, the ARTIC scheme-based Illumina COVID-seq was used, resulting in an overall genome breadth of coverage at $87.37 \pm 18.94\%$ at 1X depth across all samples. Overall, the manuscript is well structured and data analysis is well performed, the conclusions are also clearly supported by the data. There are some minor issues to address, mainly regarding methods and data visualization. Also, please check through all references to make sure the latest version is cited (e.g., ref 43).

Comments for the authors' consideration:

1. Line 55, Line 59-61, Line 78-79, and Line 100: Please add corresponding references to these sentences.

The corresponding references have been added.

2. Line 76: Check citation style.

Citation style has been fixed.

3. Line 138-141: Please provide detailed methods for the N1 and N2 RT-qPCR, e.g., reaction system volumes and amplification program.

We have added more detailed reaction and amplification information to the manuscript, as suggested (Lines 337 -345).

4. Line 141-142: Please provide detailed information regarding the N1 and N2 standard curves, i.e., slope, y-intercept and efficiency.

We have made the requested changes by adding specific calibration curve requirements to the manuscript (Lines 345 -348). We record and track the C_T of our high standard at a constant threshold every batch. This combined with our calibration curve requirements ensures quantitative consistency.

5. Line 162: Does the author have any results from the SnpEff annotation? If not, I suggest removing this sentence.

Thank you for pointing this out. Results from SnpEff annotation were used in analyzing the mutation-level data presented in the paper. SnpEff and SnpSift results are included in the summary table generated by viralrecon pipeline. This summary file contains nucleotide mutations and the resulting amino acid changes which was then compared against a manually curated list of known and epidemiologically important SARS-CoV-2 variants from outbreak.info.

6. Line 185-186: What is the average depth across all samples?

The median depth of coverage varied quite a bit across all samples (see figure below) and is now shown in Supplementary Fig. S4. We have also presented an example for the depth of coverage across the SARS-CoV-2 genome for 184 wastewater samples from one of the sequencing runs in Supplementary Fig. S5. Data on raw reads and statistics on genome coverage and depth are also presented in Supplementary Table 2.

7. Line 187: One question I have is that how the author decided lineages and the corresponding relative abundances? As the author mentioned in the discussion, there is a disagreement in the lineages identified through the consensus method (Pangolin) versus direct mapping (Freyja). Yet in this paragraph it looks like the author adopted

lineages from the Pango approach and the relative abundance from Freyja? Please clarify.

All of the downstream analysis of detection of lineages and the corresponding relative abundances described in our study were obtained through Freyja. We present Pangolin results based on consensus sequences here in the first sentence to tie it with our discussion later where we discuss that Pangolin results are not very robust in the case of wastewater samples. We have clarified this now on Line 143 (“To assess the diversity of SARS-CoV-2 lineages present in each wastewater sample, we utilized Freyja. Our analysis revealed...”).

8. Figure 190: Looks like the order of Figure 3 and Figure 2 is reversed, please check and clarify.

Thank you for pointing this out. The order of the figures has been changed as they appear in the text (now Line 146).

9. What is the "other" lineage (yellow bar) in Figure 3a, date 11-15-21? This is very eye-catching in the figure and needs to be described in some way.

Thank you for noticing this. Any other lineage except belonging to Delta and Omicron were grouped into the “Other” lineages to keep the figure simple and highlight the lineages discussed in the current study.

A large proportion of lineages on date 11-15-21 are derivatives of B.1* and B.1.1*, which we acknowledge is a bit eye-catching. We think some of these lineages might have now been updated to other Pango lineages but we chose to keep the analysis up-to-date with the information that was available at that time - early in the Omicron wave. Additionally, there were many other Pango lineages isolated from wastewater samples – way too many to fit into the plot individually. There were several lineages in the ‘proposed’ category as well as this was during the start of the Omicron wave and these lineages did not have a Pango designation. Together, all these lineages were grouped into the ‘Other’ category.

To provide more clarity, we now include an additional figure S6 in the supplementary materials (also pasted here) which breaks down some of the lineages in the ‘Other’ category and present the raw lineage data for each wastewater sample sequenced in the study in Supplementary Table 3.

10. Line 192: The 11-19 sample in Figure 3b looks like a 100% Omicron (light green bar). Please clarify.

Figure 3b (now Fig. 2C) specifically describes the proportion of Omicron sub-lineages and not all lineages. Proportion of all SARS-CoV-2 lineages are shown in figure 3a (now Fig. 2A) and Supplementary Fig. S6.

11. Line 203: Figure 4 is very busy. Consider showing only the important or mentioned lineages in the text (the ones of higher relative abundances) and keep the raw data of all Omicron lineage in the supplemental materials, e.g., a supplemental table.

We have made the requested changes to Figure 4 by grouping some of the expanded sub lineages into one. For example, lineage BA.1.1.10 has been grouped under BA.1.1* to reduce the density of points in the plot. Unfortunately, we cannot drop lineages of lower relative abundance as these are the interesting ones. Lineages generally start out at lower abundance at the start of a new wave and are the ones we discuss in the text. To provide more clarity, we have now included the raw data for all the lineages detected in our wastewater samples in the supplementary materials (Supplementary Table 3).

12. Line 222-223: This sentence needs to be clarified. It sounds like the author is talking about the pattern of BA.3 while it actually means the pattern of all Omicron sublineages.

This has been fixed. We have modified the sentence from “The patterns observed in wastewater sequencing data were reflected in clinical sequencing” to “Overall, the diversity of lineages observed in wastewater sequencing data were closely reflected in clinical sequencing” (Line 181).

13. Line 237: For Figure 5, I cannot see the details due to the low resolution. Consider better annotation some of the mutations on the y-axis, e.g., color coding these lineage-specific mutations.

Thank you for your feedback. We have only presented the Spike and ORF1ab gene mutations in the figure. We have made further changes to improve the visualization of the mutations. We have plotted the heatmaps on two rows to improve the visibility of mutations. We have also provided a high resolution figure upon final submission to enable better visibility of the mutations shown in the figure.

14. Line 240: Are all these omicron specific mutations at similar allele frequencies? The mutations had different allele frequencies (AF) but all greater than 0.25. We have modified the text in the manuscript to add this information (Line 199). Specifically, E:T9I had AF 0.52, S:G339D had AF 0.32, S:371L had AF 0.31, S:373P had AF 0.31, S:375P had AF 0.34.

15. Line 307-309 and Line 327-329: These studies used the RBD region only sequencing method. To strengthen the study's importance, I suggest the author adding in the discussion, or introduction, the advantage of sequencing whole genome in comparison to sequence a selective sub-region.

Thank you for your suggestion. We have added a few sentences about this in the discussion section (Lines 294-300).

Reviewer #3 (Comments for the Author):

1. Wastewater-based epidemiology (WBE) has become very popular and powerful as an unbiased, passive, recruitment-independent tool to monitor emergence and re-emergence of infectious diseases in post-covid era. The most recent successful example is the sustained detection of polio virus sequence in London sewage that prompted urgent vaccination advice to susceptible population. While similar studies have been reported in many cities, there are a few aspects that remain understudied. One such aspect is the effect of catchment population on WBE performance. Intuitively, a large catchment size can reduce the number of sampling points to cover the whole city population but will likely result in reduced sensitivity as the source viral RNA will become more diluted before reaching sewage treatment plants. In this study, the

catchment population size of each sewage sampling site varied widely, from as low as 1,322 to a high of 515,494. It would be interesting to know if there was any association between WBE performance with catchment population size? Were early warnings (i.e., detection of viral sequences in sewage before clinical case) more commonly contributed by less populated sampling sites? Such data would add values to the study if available and could advise on attributes that define the most cost-effective WBE design study.

Thank you for this excellent point. To address the potential effect of catchment population size on WBE performance, we performed a mixed-effects logistic regression analysis and assessed the association between normalized catchment population size (normalized by influent flow rate in mg/day), and Omicron detection (measure of WBE performance), while accounting for variability across different weeks and wastewater sampling sites. Our analysis found no statistically significant effect of normalized catchment population size on Omicron detection (Estimate = -0.0443, SE = 0.0336, p-value=0.188; Fig.1 and Table below).

Additionally, we analyzed the association between normalized catchment population size and SARS-CoV-2 lineage diversity (Shannon diversity).

Our analysis found a statistically significant negative association between the normalized population size and SARS-CoV-2 lineage diversity (Estimate = -0.0348, SE = 0.0087, p-value < 0.00; Fig.1 and Table). This suggests that when population size is higher relative to the influent flow, diversity of SARS-CoV-2 lineages tends to decrease, leading to predominance of specific viral lineages and likely an underestimation of the actual diversity of lineages circulating in the community. However, the random effects structure of the model indicates that this relationship may vary across different wastewater sites and over time.

Although the findings related to lineage diversity are noteworthy, we acknowledge that a more comprehensive dataset including additional factors influencing WBE performance would provide stronger insights. Moreover, the main focus of our current paper is on the early detection of the Omicron, and the lineage diversity analysis may deviate from this central focus. As a result, we have decided to exclude these preliminary results from the current manuscript and intend to explore these questions in a future study.

Fig.1. Relationship between catchment population size (normalized by influent flow) and detection of Omicron lineage (a), Shannon diversity (b).

Effect	Term	Estimate	Std.error	Statistic	P.value	Conf. low	Conf. high
Omicron detection as response variable							
fixed	(Intercept)	-3.299	1.542	-2.140	0.032	-6.320	-0.277
fixed	population_per_flow_mgday	-0.044	0.034	-1.317	0.188	-0.110	0.022
ran_pars	WW_site_name	0.604	-	-	-	-	-
ran_pars	Week	5.192	-	-	-	-	-
Shannon diversity as response variable							
fixed	(Intercept)	2.180	0.185	11.755	0.000	1.810	2.550
fixed	population_per_flow_mgday	-0.035	0.009	-3.995	0.000	-0.052	-0.018
ran_pars	WW_site_name	0.667	-	-	-	-	-
ran_pars	Week	0.378	-	-	-	-	-
ran_pars	Residual	0.774	-	-	-	-	-

2. Short single end 75bp mNGS was performed. Were longer contigs generated by de novo assembly before feeding to variant mapping and calling?

We did not do de novo assembly or generated contigs as we used an amplicon sequencing approach. In our study, we found that calling variants based on consensus genomes was not very robust, therefore, we performed variant calls on the reads that mapped to the reference SARS-CoV-2 genome using a threshold of at least 10 reads covering variant positions and used the Freyja pipeline to ascertain the lineages.

3. Methodological details on the collection of virus-related information from clinical settings appears to be missing.

Thank you for noticing this. We have now included details on lineage information from clinical samples in the 'Bioinformatics Analysis' section. Lines 376-379: 'To compare SARS-CoV-2 lineage diversity observed in wastewater with clinical samples, we extracted prevalence of different SARS-CoV-2 lineages from clinical surveillance in Utah using the 'outbreakinfo' R package'.

Reviewer #5 (Comments for the Author):

This is one of the most fluid and concise papers that deals with wastewater identification of viral components. This manuscript is well written and is going to add to the growing body of work surrounding SARS-CoV-2. This paper presents the results in a very readable format, and the data is presented in an effective format. The impact of this study will help public health professionals identify potential new SARS-CoV-2 variants, and help plan for potential future pandemics.

Thank you for your positive feedback and kind comments.

April 12, 2023

Dr. Pooja Gupta
Utah Department of Health
4431 South 2700 West
Salt Lake city, Utah 84129

Re: Spectrum00391-23R1 (Wastewater genomic surveillance captures early detection of Omicron in Utah)

Dear Dr. Pooja Gupta:

Your manuscript has been accepted, and I am forwarding it to the ASM Journals Department for publication. You will be notified when your proofs are ready to be viewed.

Sincerely,

Abimbola Kolawole
Editor, Microbiology Spectrum
